# Eco-evolutionary dynamics of gut phageome in wild gibbons (*Hoolock tianxing*) with seasonal diet variations

Shao-Ming Gao ®[1], Han-Lan Fei ®[1,2], Qi Li[1], Li-Ying Lan[1], Li-Nan Huang ®[1]✉ & Peng-Fei Fan ®[1]✉

It has been extensively studied that the gut microbiome provides animals flexibility to adapt to food variability. Yet, how gut phageome responds to diet variation of wild animals remains unexplored. Here, we analyze the eco-evolutionary dynamics of gut phageome in six wild gibbons (*Hoolock tianxing*) by collecting individually-resolved fresh fecal samples and parallel feeding behavior data for 15 consecutive months. Application of complementary viral and microbial metagenomics recovers 39,198 virulent and temperate phage genomes from the feces. Hierarchical cluster analyses show remarkable seasonal diet variations in gibbons. From high-fruit to high-leaf feeding period, the abundances of phage populations are seasonally fluctuated, especially driven by the increased abundance of virulent phages that kill the *Lachnospiraceae* hosts, and a decreased abundance of temperate phages that piggyback the *Bacteroidaceae* hosts. Functional profiling reveals an enrichment through horizontal gene transfers of toxin-antitoxin genes on temperate phage genomes in high-leaf season, potentially conferring benefits to their prokaryotic hosts. The phage-host ecological dynamics are driven by the coevolutionary processes which select for tail fiber and DNA primase genes on virulent and temperate phage genomes, respectively. Our results highlight complex phageome-microbiome interactions as a key feature of the gibbon gut microbial ecosystem responding to the seasonal diet.

The gut microbial community is multifunctional, impacting states of host physiology and health by participating in metabolism, immunity, and development[1]. Of these multiple roles involved, food digestion emerges as a pivotal and fundamental function of gut microbiome[2]. Alterations in diversity, structure and function of intestinal microbial flora may significantly affect the extraction of nutrition and energy and vice versa[3,4]. For most wild animals, seasonal changes in climate and geographic heterogeneity can temporally and spatially affect the availability and variety of food. Recent gene surveys and meta-omic studies have documented substantial seasonal fluctuations of the gut microbiome in response to diet variations in great ape, panda, and

gelada[5–7], shedding light on the extent to which gut microbiota plasticity provides dietary and metabolic flexibility to the wild animals[8].

As an important and abundant component of the gut microbial ecosystem, viruses (mainly phages) markedly impact their prokaryotic hosts dynamics via lytic infection[9], enhance bacterial fitness with auxiliary metabolic genes (AMGs)[10], and contribute to phage-host coevolution through antagonistic interactions[11]. Therefore, gut phages might also play important roles in the process of animal adaptation to diet variations by shaping the eco-evolutionary dynamics of their prokaryotic hosts. However, the ecological study of gut phageome has considerably lagged behind their prokaryotic counterparts largely due

[1]School of Life Sciences, Sun Yat-sen University, Guangzhou 510275, PR China. [2]College of Life Science, China West Normal University, Nanchong 637002, PR China. ✉e-mail: eseshln@mail.sysu.edu.cn; fanpf@mail.sysu.edu.cn

to the experimental and methodological limitations[12]. While recent investigations combining metagenomic sequencing and other methods have clarified aspects of gut phage diversity and associated impacting factors, e.g., age, geographic location, disease state, and individuality[13–16], the relationships between gut phageome and animal diet have rarely been characterized. To date, several studies have examined phageomes in human, murine, and giant panda guts, revealing patterns whereby viral diversity or abundance were sensitive to dietary disturbances[17–22]. A few other gut viromic analyses have revealed a repository of beneficial genes involved in energy harvest such as carbohydrate and amino acid metabolism, suggesting a potential role of metabolic reprogramming by gut phages[10,23,24]. These pioneering works have provided initial insights into the ecological linkages between diet and the gut phage community. Nevertheless, the potential influence of dietary variations on gut phageome in wild animals remains unexplored.

The investigation of gut phageome dynamics in wild animals possibly associated with dietary variations faces multiple challenges, especially in quantifying diet and collecting fresh feces of individually recognized animals for consecutive days[25]. Specifically, the elusive nature of many wild animals (e.g., predators, ungulates, etc.) prohibits habituation, individual identification, direct observation of feeding behavior, and collection of fresh feces upon defecation. Consequently, recent studies typically collected and analyzed fecal samples left behind in the field by groups of wild great apes, with the individual origin of fecal source and elapsed time between defecation and sampling being unknown[6]. To quantify diet composition, researchers adopted a DNA metabarcoding on the fecal samples, or even simply used climate as a proxy[5,6]. While the former method may provide information about the plant species present in the feces, it fails to resolve which specific parts (e.g., fruits vs. leaves) of the plants are consumed by the animals, and at what proportions.

In this work, we overcome these limitations by employing an uncommon strategy in which time series and full-day feeding behavior data were recorded and parallel fresh feces were collected for six habituated and individually recognized skywalker hoolock gibbons (*H. tianxing*) from two family groups (NK: Nankang, BC: Banchang) residing in Mt. Gaoligong, Yunnan, China in 15 months. A recent microbiome study has showed that the gut prokaryotes of these gibbons responded to the seasonal diet compositionally and functionally[25]. In this study, we focus specifically on the eco-evolutionary dynamics of gut phageomes by analyzing a massive longitudinal dataset of viral metagenomes (VMs) and microbial metagenomes (MMs) for the six gibbons. As phages depend on their hosts to replicate, we hypothesize that the gibbon diet variations would result in an apparent seasonal pattern of gut phages driven by their prokaryotic hosts' dynamics. Given that virulent phages (replicating exclusively through a lytic cycle) and temperate phages (replicating either through a lytic or a lysogenic cycle) exhibit distinct co-replicative states with their hosts[26], we further hypothesize that the response of phageome to diet variations in the gibbons would be divergent between phages with different lifestyles. Our analyses advance the understanding of the dynamic gut microbial ecosystem key for the adaptation of wild animals to seasonal diet.

## Results
### A genomic catalog of gut DNA phages from the wild gibbons
To explore the diversity and temporal variability of gut phages in wild gibbons, we collected 139 fecal samples with matched full-day dietary data of six skywalker hoolock gibbon individuals (Fig. 1a). A total of 145 VMs (125 samples, of which ten were randomly selected and subjected to 3-replicates multiple replacement amplification) and 138 MMs were successfully generated (Fig. 1b and Supplementary Data 1). After quality filtering, the VMs and MMs yielded an average of 46,016,884 and 96,032,291 paired reads per library, respectively (Supplementary

Fig. 1). Overall, VMs displayed a significantly higher phage enrichment efficiency, as evidenced by fewer reads aligned to small- and large-subunit ribosomal RNA genes, and single-copy universal markers for bacterial and archaeal ribosomal proteins (Supplementary Fig. 2).

A total of 17,449 and 21,749 single-contig phage genomes were recovered from the 145 VMs and 138 MMs, respectively (Fig. 1b). These phage sequences ranged between 10 and 370 kb (with ~89.9% from 10 to 50 kb) in size, and both VMs and MMs recovered phage genomes larger than 360 kb (Fig. 1c, Supplementary Data 2). Specifically, the number of phage genomes from VMs was comparable to that of phage genomes from MMs in different genome length ranges (Fig. 1c). To resolve the composition of recovered phage genomes, we used the graph convolutional network method which allowed taxonomic assignment for 83.0% of phage genomes to 18 families of double-stranded DNA (dsDNA) viruses (Fig. 1d). The majority of taxonomically classified phage genomes from both method profiles were assigned to *Salasmaviridae*, *Peduoviridae*, *Casjensviridae*, *Autographiviridae*, *Mesyanzhinovviridae*, and *Chaseviridae*. While all classified phage families were found in both VMs and MMs, further inspection of the phage genomes revealed that MMs yielded higher genome completeness (estimated based on average amino acid identity in CheckV)[27], and a higher proportion of prophages than VMs (Fig. 1d). Across the gibbon individuals, with B2 as the only exception, the number of identified phage genomes in MMs were significantly higher than that in VMs (Fig. 1e). Together, these results show that the MMs could recover a large and comparable number of phage genomes to VMs while this approach displayed more signatures from cellular organisms.

### Distinct landscapes of phage populations captured in the VMs and MMs
All 39,198 recovered phage genomes were clustered into a set of 2789 phage populations (vOTUs) (Fig. 2a), each of which was then split into 1571 VM-vOTUs and 2152 MM-vOTUs based on the source of its members within the population (Fig. 2a, Supplementary Data 3, 4). In total, 637 vOTUs were exclusively found in VMs (defined as VM-specific vOTUs), 934 in both VMs and MMs (shared vOTUs), and 1218 in MMs alone (MM-specific vOTUs) (Fig. 2a). Representative genomes of VM-vOTUs and MM-vOTUs were selected and aligned to the VM and MM reads per sample, respectively (Supplementary Figs. 3, 4). When averaged at the sample level, both the mean relative abundance (VM-shared: 0.064%, MM-shared: 0.067%, VM-specific: 0.063%, MM-specific: 0.030%) and the prevalence (VM-shared: 64.7%, MM-shared: 56.4%, VM-specific: 58.8%, MM-specific: 51.1%) were significantly higher for the shared vOTUs than the specific vOTUs (Fig. 2b, c), suggesting that only dominant phage populations could be detected by both metagenomic approaches. Moreover, the number of MM-specific vOTUs were significantly higher than that of shared vOTUs (Fig. 2d). As a result, either phage richness or normalized abundance profiled through the two approaches were nonsignificantly or negatively correlated (Fig. 2e, f). However, except in gibbon A1, phage community dissimilarity derived from VMs and MMs were significantly correlated in other five gibbon individuals (Supplementary Fig. 5). Our results also showed that 93.9% of the VM-vOTUs while 75.4% MM-vOTUs could be detected in both gibbon family groups (Supplementary Fig. 6a). Further analyses revealed that VM-vOTUs were more prevalent than the MM-vOTUs (Supplementary Fig. 6b), suggesting that VM-vOTUs might be more diffusible across the gibbons.

We subsequently merged all phage genomes recovered by both approaches for further analysis. Since MMs captured higher proportion of prophages than VMs (Fig. 1d), we distinguished the 39,198 predicted phage genomes into 15,495 virulent and 23,703 temperate phage-derived sequences. Further inspection revealed that VMs detected higher proportion of virulent phages and lower proportion of temperate phages than MMs (Fig. 2g, Supplementary Fig. 7). Thus, the

2789 clustered phage populations were distinguished into 1072 virulent (Vir-vOTUs) and 2,157 temperate (Tem-vOTUs) phage populations based on the predicted lifestyle of its members within the population (Fig. 2h). Notably, 440 vOTUs, each of which contained both virulent and temperate phage genomes, were split into Vir-vOTUs and Tem-vOTUs (Fig. 2h). Among the predicted virulent genomes were 195 putative crAss-like phages which were further clustered into three Vir-vOTUs (Supplementary Fig. 8). These genomes displayed low GC content (mean = 37.7%), extensive usage of an alternative genetic code (code 15), and a maximum length of ~115 kb, supporting previous findings that crAss-like phages are obligate lytic and contain several unusual features[28]. Representative genomes of Vir-vOTUs and Tem-vOTUs were separately aligned to reads of VMs and MMs, respectively (Supplementary Data 5, 6). The number of Tem-vOTUs was significantly higher than that of Vir-vOTUs in all fecal samples (Fig. 2i).

Together, these results indicated that VMs and MMs recovered distinct landscapes of phage populations which biased towards virulent and temperate types, respectively.

## Phage community structure and responses to seasonal diet of gibbons

We next investigated phage community structure and their dynamics in response to gibbon diet variations. We performed separate principal coordinates analysis (PCoA) based on Bray-Curtis dissimilarity and the Permutational multivariate analysis of variance (PERMANOVA) analysis, which revealed highly significant differences in both virulent ($R^2 = 0.42$) and temperate ($R^2 = 0.63$) phage communities between the gibbon family groups (Fig. 3a). As most of the virulent (96.5%) or temperate (98.1%) phage populations could be identified in the metagenomes of gibbons A2 and B2 and only a limited number of

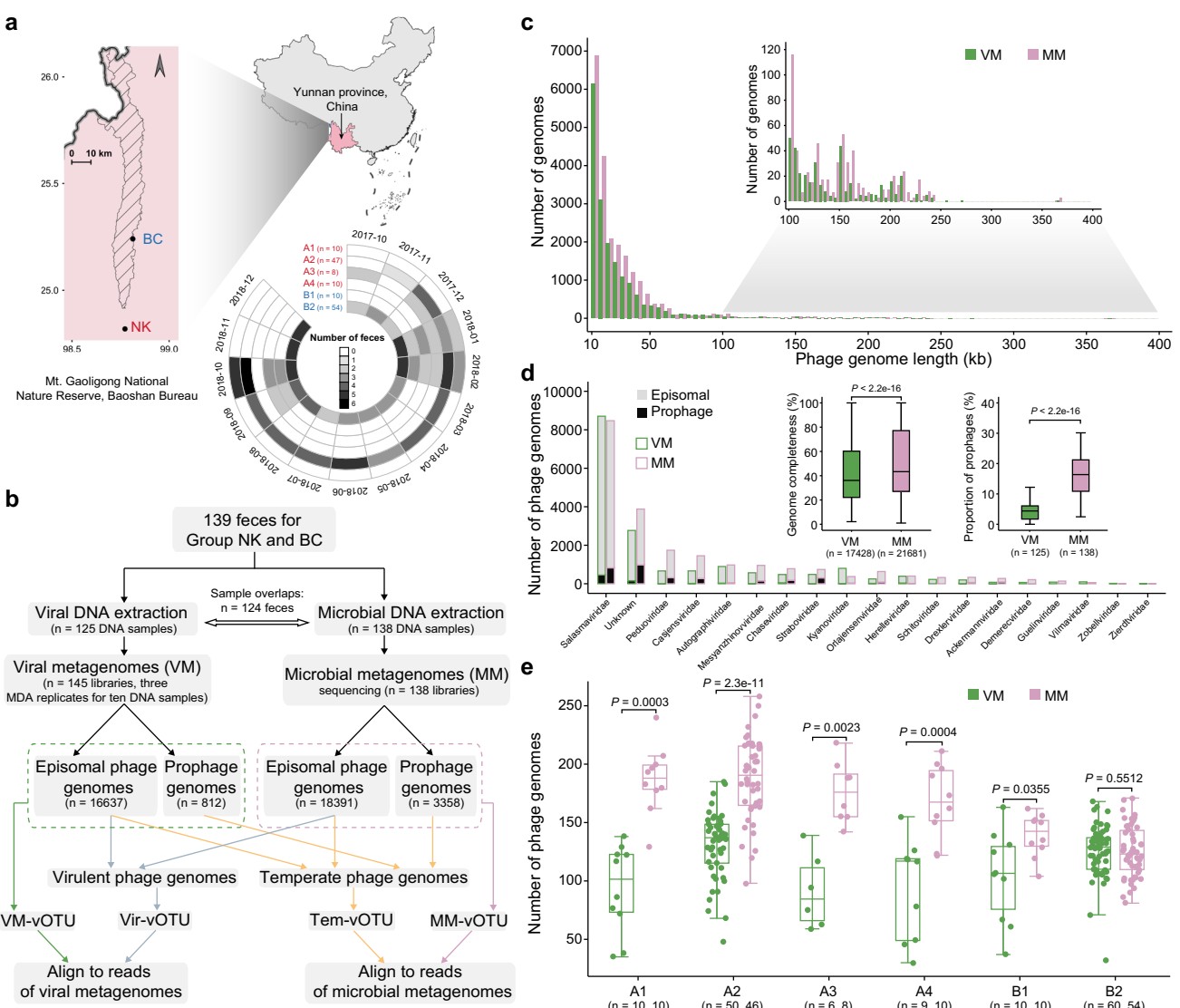

**Fig. 1 | Experimental design and the performance of VMs and MMs. a** Locations of the two family groups of wild and female skywalker hoolock gibbons (group NK in red and NK in blue) and the number of associated fecal samples per month from October 2017 to December 2018. China boundary was obtained from the National Catalog Service for Geographic Information. **b** Overview of the experimental methods and bioinformatic pipelines used for VMs and MMs. **c** Distribution of phage genome size colored by the two metagenomic approaches (VM: green, MM: purple). **d** Distribution of phage taxonomic families recovered by the two metagenomic approaches. Inset: Comparison of the phage genome completeness (left panel) and the proportion of prophages (right panel) between the two approaches. The prophage proportion for the three replicates of ten randomly amplified VM samples were averaged. **e** The number of phage genomes recovered by the two different metagenomic approaches across fecal samples of the six gibbons. In boxplots, boxes represent the interquartile range (IQR), and the lines inside show the median. The lower and upper whiskers correspond to the lowest and highest values within 1.5 times the IQR. Statistical significance is based on non-parametric Wilcoxon t-test (unpaired and two-sided), and the n number (i.e., the sample size used to derive statistics) are provided for each group. Source data are provided as a Source Data file.

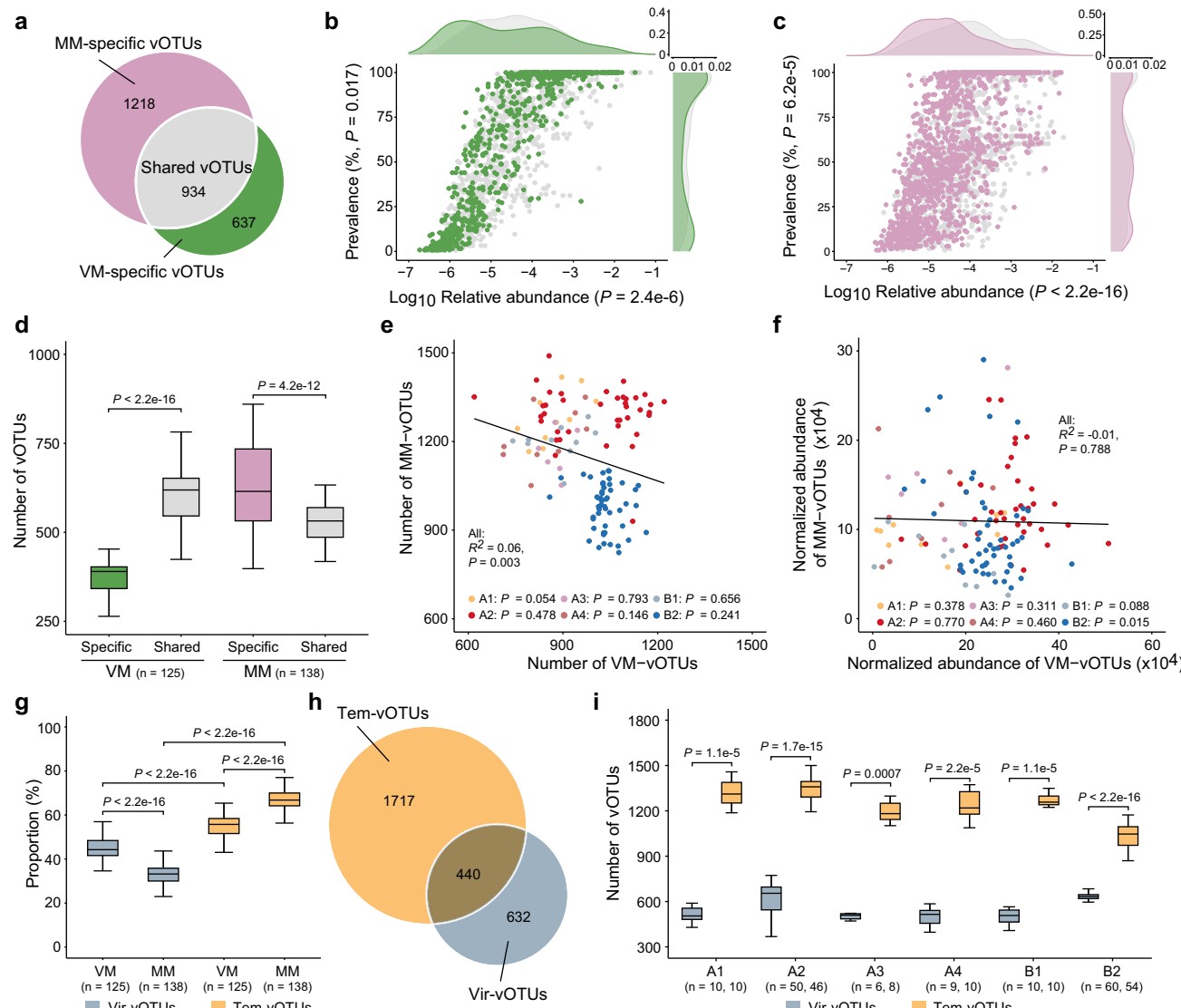

**Fig. 2 | Comparison of gibbon gut phageomes recovered through VMs and MMs. a** Venn diagram shows the number of specific and shared viral populations detected by the two approaches. The relative abundance (x-axis) and prevalence (y-axis) of specific and shared phage populations detected from VMs (**b**) and MMs (**c**). *P*-values in brackets indicate the statistical significance of the relative abundance or the prevalence between groups based on non-parametric Wilcoxon *t* test (unpaired). The density plots parallel to the axes show the distribution of the prevalence and relative abundance. The VM-specific vOTUs, shared vOTUs, and MM-specific vOTUs are colored in green, gray, and purple, respectively. **d** The number of specific and shared VM-vOTUs in VMs, and the number of specific and shared MM-vOTUs in MMs. Correlations between the number (**e**) or the normalized abundance (**f**) of phage populations recovered from VMs and MMs colored by the six gibbons. The adjusted $R^2$ values and best-fit lines for the linear regressions are

presented. **g** Proportion of the virulent or temperate phage populations recovered from VMs and MMs in each sample. **h** Venn diagram shows the number of phage populations with different lifestyles. The overlapped section indicate phage populations that included both virulent and temperate types and were thus split into Vir-vOTUs and Tem-vOTUs. **i** The number of virulent and temperate viral populations in each fecal sample of the six gibbon individuals. In boxplots, boxes represent the interquartile range (IQR), and the lines inside show the median. The lower and upper whiskers correspond to the lowest and highest values within 1.5 times the IQR. Statistical significance is based on non-parametric Wilcoxon *t* test (unpaired and two-sided), and the n number (i.e., the sample size used to derive statistics) are provided for each group. Source data are provided as a Source Data file.

phage populations were specifically found in gibbons A1, A3, A4, and B1 (Fig. 3b), we subsequently focused on the phage population dynamics of gibbons A2 and B2 to resolve the variations of virulent and temperate phage assemblages with gibbon diet variations.

Based on the observed feeding time each gibbon spent on specific food types (i.e., fruit, leaf, flower, animal, and others), we could separate the dietary composition of each gibbon into high-fruit (HF, mean fruit proportion = 83.3%) feeding season and high-leaf (HL, mean leaf proportion = 75.0%) feeding season through hierarchical clustering (Supplementary Fig. 9). We firstly conducted non-parametric test for the abundance of each virulent or temperate populations between the

two dietary seasons. Significant shifts were found in the abundance of 12 Vir-vOTUs and 152 Tem-vOTUs in gibbon A2, and 68 Vir-vOTUs and 103 Tem-vOTUs in B2; these phage populations were thus defined as diet-responsive vOTUs (Fig. 3c, Supplementary Data 7). Across these diet-responsive vOTUs, only one Vir-vOTU and 16 Tem-vOTUs were shared between gibbons A2 and B2, emphasizing the individuality of phage communities between the two gibbons (Fig. 3c). We also predicted potential prokaryotic hosts for virulent (41.4% of Vir-vOTUs) and temperate (70.8% of Tem-vOTUs) phage populations, allowing for the investigation of phage-host dynamics (Supplementary Data 8). For the identified diet-responsive vOTUs, the abundance of six

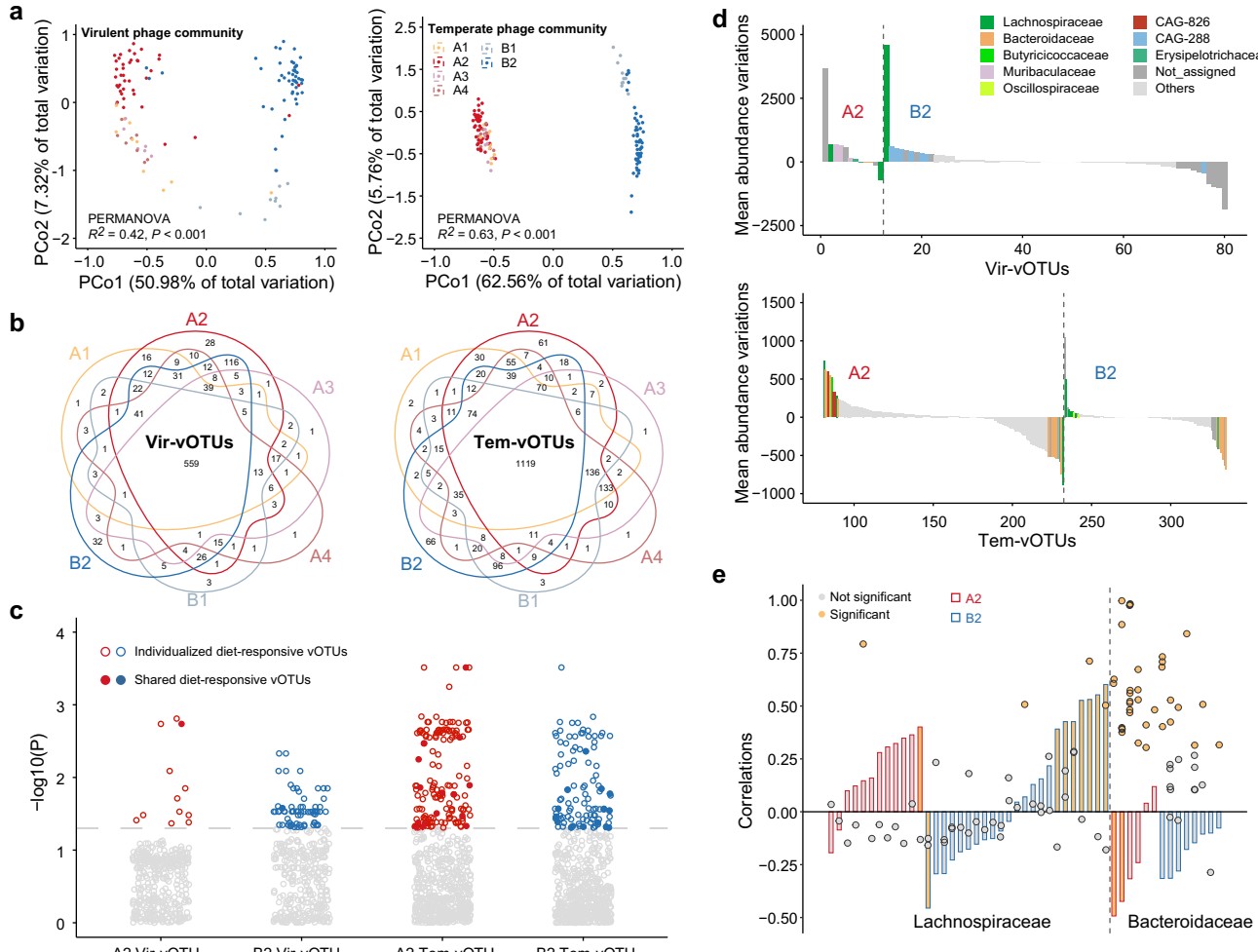

**Fig. 3 | Phage community variation across gibbons and their feeding seasons.**
**a** Principal coordinates analysis (PCoA) of phage community structure (using Bray-Curtis dissimilarity) for the six gibbons after separation into virulent (left panel) and temperate (right panel) fractions. The PERMANOVA consider samples grouped by gibbon families. **b** Venn diagram shows the number of virulent (left panel) and temperate (right panel) phage populations in the six gibbons. **c** Manhattan plots show the virulent and temperate populations with abundance significantly changed between the HF and HL seasons. Solid dots represent diet-responsive phage populations identified in both gibbons A2 and B2, while the hollow dots represent diet-responsive phage populations specifically found in A2 or B2. **d** Bar graphs indicate variations in the mean abundance of diet-response virulent (top panel) and temperate (bottom panel) phage populations colored by host families from HF to HL seasons. The variation value was calculated by subtracting the mean abundance of phage populations in HF season from that in HL season. Only populations mostly (top ten) varied in abundance are highlighted. **e** The x axis represents hosts of the six *Lachnospiraceae*-associated Vir-vOTUs significantly increased in abundance and the 30 *Bacteroidaceae*-associated Tem-vOTUs significantly decreased in abundance from HF to HL season. Barplots indicate the Pearson's correlations between the abundance of these host populations and leaf proportion in gibbons A2 and B2. Dot plots indicate the Pearson's correlations between the abundance of *Lachnospiraceae* populations and corresponding virulent populations, and the abundance of *Bacteroidaceae* populations and corresponding temperate populations. *P*-values for multiple testing were adjusted using the Benjamini and Hochberg false discovery rate controlling procedure. Source data are provided as a Source Data file.

*Lachnospiraceae*-associated Vir-vOTUs (1 in gibbon A2 and 5 in B2) and 30 *Bacteroidaceae*-associated Tem-vOTUs (18 in A2 and 12 in B2) were mostly increased or decreased from HF to HL, respectively (Fig. 3d). Similar trends were observed at the community level in which the *Lachnospiraceae*-associated virulent phages and the *Bacteroidaceae*-associated temperate phages contributed most to the increase or decrease of total phage abundance with diet variation of gibbons (Supplementary Fig. 10).

Since the phages might impact their host abundances, we then investigated whether the temporal trends of these diet-responsive phages matched those of their putative hosts. We specifically focused on the above identified six *Lachnospiraceae*-associated Vir-vOTUs and 30 *Bacteroidaceae*-associated Tem-vOTUs. These virulent and temperate phages were linked to 35 *Lachnospiraceae* populations and 14 *Bacteroidaceae* populations, respectively (Fig. 3e). Among them, eight *Lachnospiraceae* populations and two *Bacteroidaceae* populations

significantly increased and decreased their abundances with leaf proportion, respectively, paralleling the abundance trajectories observed in their phages (Fig. 3e). Meanwhile, significant phage-host abundance correlations were found for two *Lachnospiraceae* populations and two *Bacteroidaceae* populations responsive to leaf proportion, implying a scenario under which the virulent phages increased in abundance by culling but not necessarily eradicating the *Lachnospiraceae*, while the temperate phages piggybacked the *Bacteroidaceae* hosts and thus decreased in abundance. Altogether, these results provided evidence for our hypothesis that virulent and temperate phages displayed distinct responses to the dietary changes in gibbons.

**Phage functional plasticity across lifestyles and dietary seasons**
To further resolve functional diversity of the gibbon gut phages, we clustered all predicted protein-coding genes from the virulent phages (593,609 genes), temperate phages (774,454), and host genomes

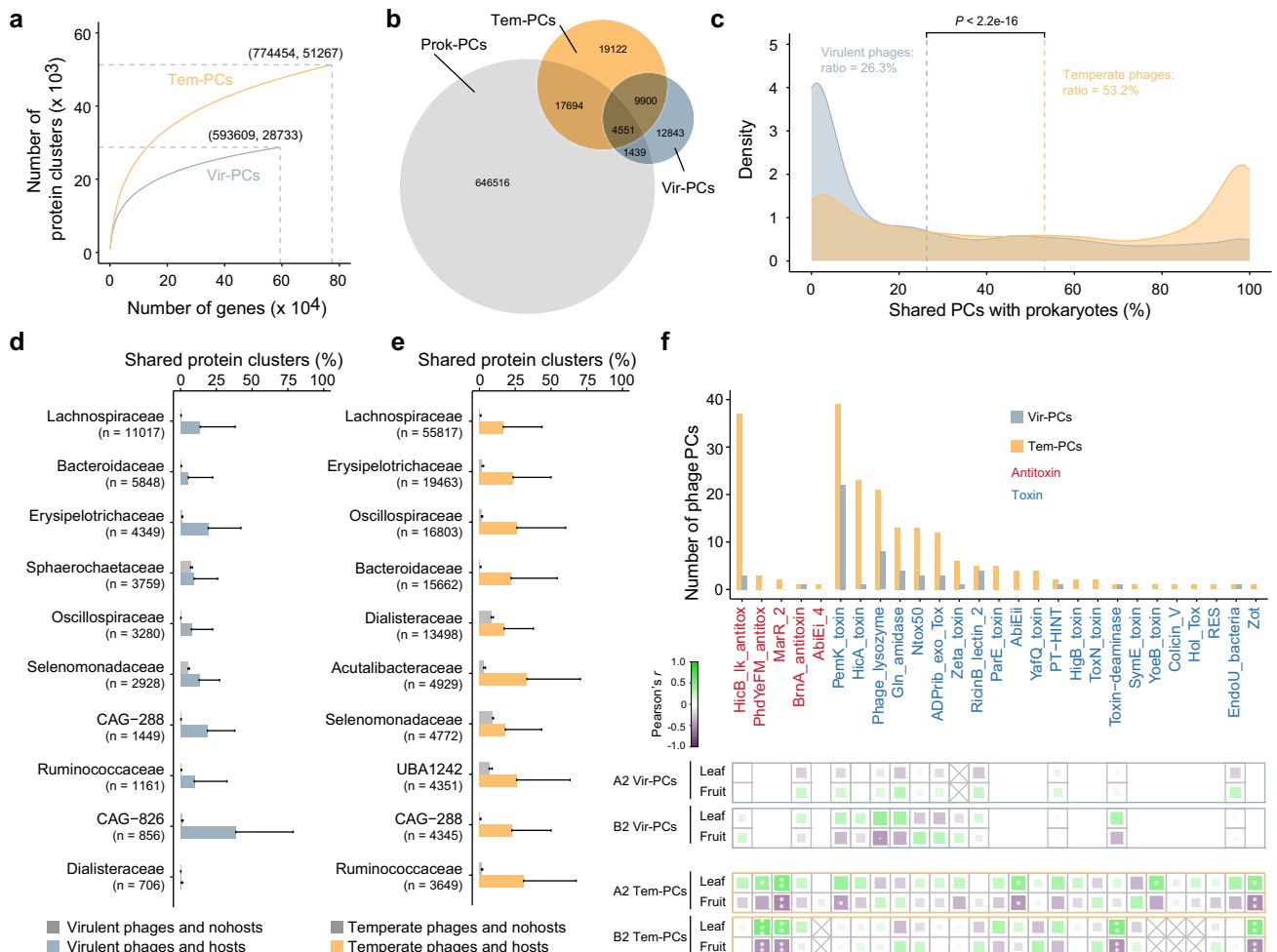

**Fig. 4 | Phage functional diversity and response to season diet. a** Accumulation curves of virulent and temperate phage PCs. **b** Venn diagram shows the number of PCs from virulent phages (Vir-PCs), temperate phages (Tem-PCs), and prokaryotic genomes (Prok-PCs). **c** Distribution of the ratio of shared PCs with prokaryotes in virulent and temperate genomes. Dashed lines indicate the average ratio in the distribution. Statistical significance is based on non-parametric Wilcoxon *t* test (unpaired). Barplots show the proportion of shared PCs between pairs of virulent phages (**d**) or temperate phages (**e**) and prokaryotes. The blue and orange bars represent the proportion of shared PCs for virulent and temperate phages and their predicted hosts in each host family, respectively. The gray bars represent the proportion of shared PCs for virulent and temperate phages and the nonhosts in

each family. The error bar represent the standard deviation of the proportions. The n number in each parenthesis indicate the number of phage-host pairs in each family. Host families with most (top ten) number of phage-host pairs are shown. **f** The toxin-antitoxin genes identified on phage genomes and their abundance variations with dietary changes. Barplots show the number of PCs annotated as toxin and antitoxin genes found on virulent and temperate genomes. Color gradient in the heatmap denotes the Pearson's correlations between the abundance of PCs related to toxin/antitoxin genes and dietary proportions in gibbons A2 and B2. *P* values for multiple testing were adjusted using the Benjamini and Hochberg false discovery rate controlling procedure. Source data are provided as a Source Data file.

(17,780,751, with genes from co-binned phages and prophages removed) into non-redundant protein clusters (PCs). Finally, 28,733 virulent phage PCs (Vir-PCs), 51,267 temperate phage PCs (Tem-PCs), and 670,200 prokaryotic PCs (Prok-PCs) were obtained (Fig. 4a, b). We defined a PC containing both phage genes and prokaryotic genes as a shared PC. Accordingly, 20.8% of the Vir-PCs and 43.4% of the Tem-PCs were identified as shared PCs (Fig. 4b). For both virulent and temperate phages, the sizes of genes in the shared PCs were significantly larger than those in the unshared PCs (Supplementary Fig. 11). It is also notable that temperate phages contained significantly higher proportions of shared PCs than virulent phages across the identified phage genomes (Fig. 4c), implying a higher frequency of genetic exchange between temperate phages and their hosts.

In addition, we calculated the proportion of shared PCs between pairs of phages and prokaryotes. The results showed that either the virulent or the temperate phages shared a higher proportion of PCs with their hosts than with the nonhosts in all prokaryotic families (Fig. 4d, e), suggesting that phages were more likely to transfer genes

with their hosts. When averaging at the prokaryotic family level, lineage-specific shared PCs ratios were highest in CAG-508, UBA660, *Oscillospiraceae*, and UBA1242 (Supplementary Fig. 12a). Further, increased ratio of shared PCs was significantly associated with increase in the number of prokaryotic populations and PCs across the families (Supplementary Fig. 12a). Meanwhile, we found a significant and positive correlation between the total number of PCs and the ratio of shared PCs across prokaryotic genomes (Supplementary Fig. 12b). These results indicated the potential roles of shared PCs in the taxonomic and functional diversification of prokaryotes.

Among the shared PCs, 20 Vir-PCs and 131 Tem-PCs were annotated as genes related to toxin/antitoxin (TA) systems. Further screening of the recovered phage genomes identified a total of 53 Vir-PCs and 203 Tem-PCs related to these TA genes (Fig. 4f, Supplementary Data 9). For both the virulent and temperate phages, Pemk and HicB were the most annotated toxin and antitoxin proteins, respectively (Fig. 4f). The abundance of five toxin genes (Pemk, AbiEii, Toxin-deaminase, YoeB, and Zot) and two antitoxin genes (PhdYeFM and

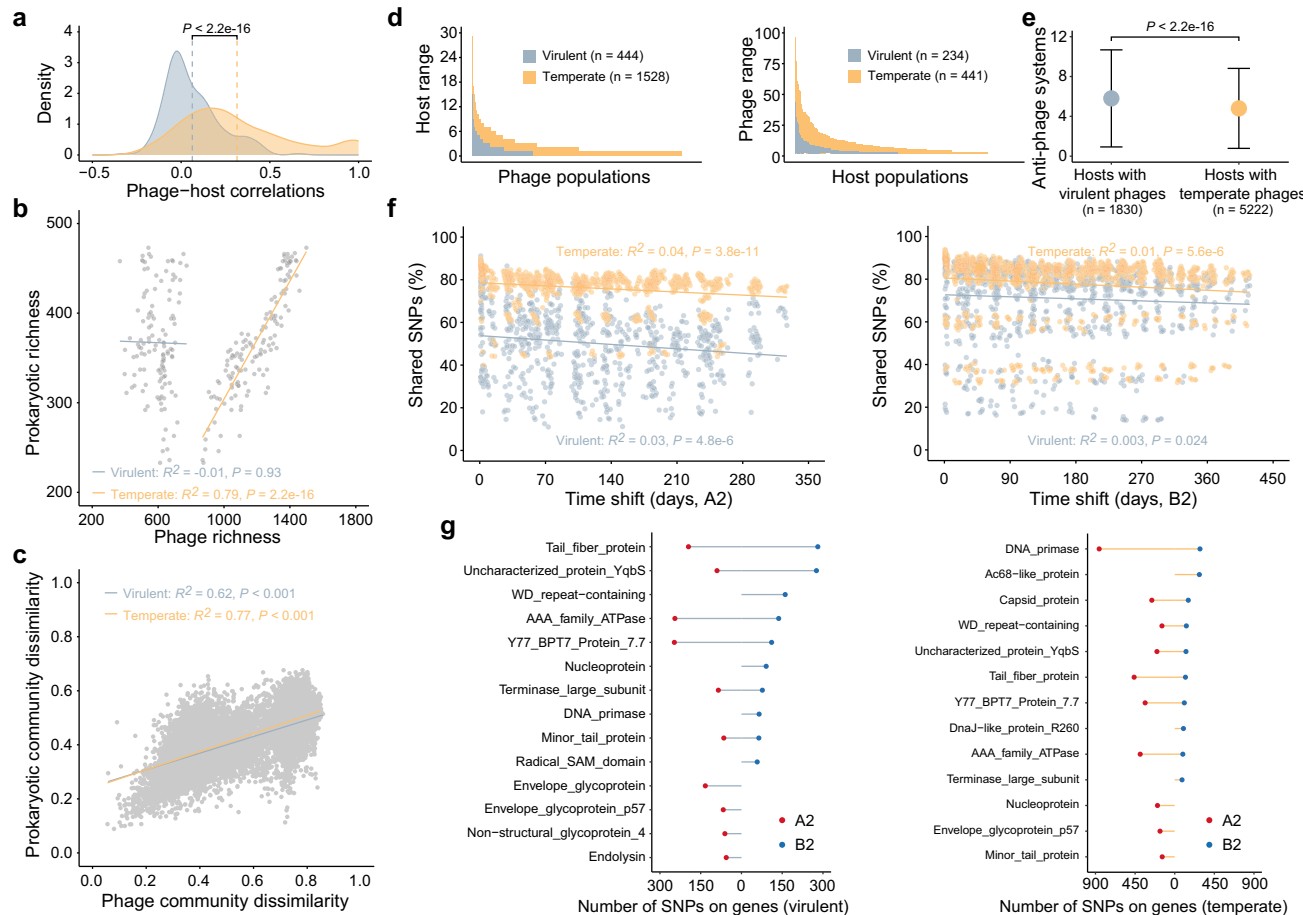

**Fig. 5 | Phage-host interactions between lifestyles and across the sampling dates. a** Distribution of virus-host correlations across the fecal samples of the six gibbons. Dashed lines indicate the average correlation coefficients in the distribution. **b** Correlations between the number of virulent or temperate populations and the number of prokaryotic populations across the samples of the six gibbons. The adjusted $R^2$ values and best-fit lines for the linear regressions are presented. **c** Mantel's correlation coefficients used to represent the associations between Bray-Curtis dissimilarities of virulent or temperate phage community and prokaryotic community. **d** Comparison of the host range (left panel) and phage range (right panel) by phage lifestyles. **e** The number of anti-phage systems in genomes of prokaryotes assigned to virulent and temperate phages. Dots represent mean number and segments indicate the standard deviation. **f** The proportion of shared SNPs identified by comparing the SNPs on virulent and temperate phage genomes at each date with all subsequent dates in gibbons A2 (left panel) and B2 (right panel). The adjusted $R^2$ values and best-fit lines for the linear regressions are presented. **g** Detailed functional information of annotated genes with most (top ten) number of SNPs located on virulent (left panel) and temperate (right panel) phage genomes in gibbons A2 and B2. Statistical significance is based on nonparametric Wilcoxon $t$ test (unpaired and two-sided), and the n number (i.e., the sample size used to derive statistics) are provided for each group. Source data are provided as a Source Data file.

MarR_2) on temperate phage genomes (both episomal and prophage) significantly increased with leaf proportion or decreased with fruit proportion in gibbons A2 and B2 (Fig. 4f, Supplementary Data 9). Meanwhile, we observed a significantly increased abundance of shared PCs containing these diet-responsive TA genes with increased leaf proportion (Supplementary Fig. 13), indicating that changes in the abundance of TA genes were mainly attributed to those in the shared PCs. In contrast, we only observed a decrease in the abundance of phage lysozyme on virulent phage genomes in HF for gibbon B2. These results suggested that temperate phage-encoded TA genes especially those in shared PCs were responsive to gibbon diet variations.

### Phage-host interactions and eco-evolutionary dynamics

As prokaryotic hosts substantially impacted both the virulent and temperate phages, we further investigated the phage-host interactions. We firstly examined the one-to-one correlation between the abundance of each phage population and its host across the fecal samples. Results showed that temperate phage populations displayed significantly stronger correlations with their hosts (Pearson's $r = 0.31$ on average) than the virulent phage populations (Pearson's $r = 0.06$ on

average) (Fig. 5a). Furthermore, the temperate phages were more closely related to prokaryotes than virulent phages in both richness and community dissimilarity (Fig. 5b, c).

We next investigated whether the phage-host interaction structure might corresponded to and impact the observed phage-host correlations. We found significantly higher host range (number of host populations for each phage population) for temperate phages, and significantly higher phage range (number of phage populations for each prokaryotic population) for hosts associated with temperate phages (Fig. 5d), suggesting that virulent phages tended to be specialists while temperate phages were more likely to be generalists. To verify this, we firstly screened the prokaryotic metagenomic-assembled genomes (MAGs) recovered from the 138 MMs for genomic features to predict the anti-phage systems. The restriction-modification (RM), followed by CRISPR-Cas, were the two most prevalent systems in prokaryotic genomes (Supplementary Fig. 14). With the increase in the number of anti-phage systems in prokaryotic genomes, the number of temperate phages infecting the prokaryotes increased faster than that of virulent phages (Supplementary Fig. 14). Meanwhile, the number of anti-phage systems was significantly higher

in the prokaryotic genomes assigned with virulent phages than in those assigned with temperate phages (Fig. 5e). These results suggested that the prokaryotes displayed a stronger and broader resistance to virulent phages than to temperate phages.

Finally, phage population genomes with a minimum sequencing depth of 10-fold across all samples of each gibbon were selected (Supplementary Fig. 15), and aligned to metagenomic reads to call single-nucleotide polymorphisms (SNPs) that might indicate relics of phage adaptation to their host. Results showed that phage genome similarity (the percentage of shared SNPs on different dates) decreased as a function of the time lag between dates, irrespective of the phage lifestyles and the reference time point (Fig. 5f). Moreover, the percentage of shared SNPs was significantly lower for virulent phages than temperate phages across the time lags, indicating a more frequent turnover of intrapopulation variants for virulent phages (Fig. 5f, Supplementary Fig. 16). Further evaluation showed that more than half of the SNPs were categorized as synonymous substitutions (Supplementary Fig. 17). As the majority of the phage genes were functionally unannotated, we thus examined the genes with annotated functions (Supplementary Data 10). We noted strong biases in the location of SNPs-targeted regions in which most genes on virulent and temperate genomes encoded tail fiber protein and DNA primase, respectively (Fig. 5g).

## Discussion

While wild gibbons are typically frugivorous[29], our study groups, which represent the northernmost margin of global gibbon species distribution[29], experience drastic seasonal fluctuations in dietary composition and could shift to a diet mainly of leaves when fruits are not available[30]. Thus, these gibbons are uniquely suited for exploring potential linkages between seasonal diet variations, gut microbiome and phageome of wild animals.

Comparative analysis of the full-sets of gibbon fecal metagenomes generated by the two approaches (VMs and MMs) recovered comparable numbers but complementary compositions of phage genomes. While VMs have been demonstrated superior in capturing rare virosphere[31], a vast number of phage genomes were recovered in our MMs. This result is reasonable as lysogeny is widely distributed in animal gut[32], and the deeper sequencing depth may have compensated the underrepresence of phage genomes in the MMs (where prokaryotic genomes dominated). Subsequent analyses further illustrated that temperate phages were more frequently detected in the MMs, while the VMs significantly enriched diffusible and virulent phages across the gibbons. This distinctive composition could stem from methodological limitations associated with phageome DNA preparations[33–35]. Phage lineages that were previously reported to be prevalent in the animal guts, e.g., *Microviridae* with single-stranded DNA genomes[36], and Lak phages infecting the *Prevotella*[37], were not recovered in our dataset. This notable difference highlights potential biases associated the multi-omics approaches and respective bioinformatics used, making comparison across studies challenging. Nevertheless, our results supported that an integrated application of the two complementary approaches would allow more comprehensively capturing the diversity of phage populations[38], and a better understanding of their ecological dynamics in the igibbon guts.

Despite the apparent separation of phage community structure between the gibbon family groups, our analyses revealed that virulent and temperate phages displayed significantly divergent responses to the seasonal diet variations. Our results are corroborated with a previous study which declared that primate phageomes within species are structured by the animal superhost[39]. Consistent with this scenario, the diet-responsive phage populations are largely gibbon individual-specific. However, such personalized responses of phage populations do not extend to higher prokaryotic-host taxonomic levels as we observed similar patterns in the abundances of *Lachnospiraceae*-

associated virulent phages and *Bacteroidaceae*-associated temperate phages in both gibbons (A2 and B2). Members of *Lachnospiraceae* are known to be fibrolytic, and have been reproducibly observed across animal species[5,40], including our recent survey of the gut prokaryotes in the six gibbon individuals examined in this study[25]. The simultaneous enrichment of *Lachnospiraceae* and its virulent phages in the HF season does not support the Kill-the-Winner dynamics which postulates substantial virulent phage-driven reduction of dominant host populations. Therefore, we suggest that the virulent phages may adopt a Cull-the-Winner strategy[41], preying on the leaf-responsive *Lachnospiraceae* species but not completely killing them towards extinction. On the other side, species of *Bacteroidaceae* are commonly found in the gut with carbohydrate-rich diets, capable of digesting non-cellulosic polysaccharides and soluble sugars as engery sources[42,43]. Thus, it is somewhat expected that negative correlations between the abundance of *Bacteroidaceae* populations and leaf proportion were found in our data sets. While temperate phages could also be induced and enter the lytic cycle whereby new progenies are released following host cell lysis[26], the depletion of *Bacteroidaceae* populations with leaf proportion was unlikely a result of prophage induction, since parallel decreases in the abundance of *Bacteroidaceae* populations and associated temperate phages were observed. Therefore, our results hinted at a Piggyback-the-Winner dynamic for *Bacteroidaceae* populations and their temperate phages which favor lysogenic infections.

We observed that phages with two different lifestyles displayed appreciably distinct modes of shared gene content with their hosts. This finding is consistent with the mosaic theory which states that phage genomes are composed of genetically diverse genes originated from divergent sources[44]. Previous studies have proposed that the frequency of horizontal gene transfers (HGTs) within phages varies with their lifestyles[45]. However, the extent and modes of HGTs between phages and their hosts remain unresolved, although a series of studies have provided evidence of phage-encoded AMGs derived from their prokaryotic hosts[46,47]. Here we showed that temperate phages exhibit higher genetic homology with their hosts compared with virulent phages, implying more frequent gene exchanges on temperate genomes and their profound implications on HGTs in the gut. We acknowledge that more rigorous approaches, e.g., phylogenetic and best-match analysis, should be used to identify HGTs, but such approaches are either tree-based or require reference genomes and thus not suitable for phage community-wide analysis[48]. Furthermore, our results also revealed that temperate phages typically enriched TA genes (e.g., PhdYeFM, MarR_2, PemK, AbiEii, Toxin-deaminase, YoeB, and Zot) in response to the seasonal diet of gibbons. TA systems are widely distributed in prokaryotes and mobile genetic elements (MGEs) that are prone to HGT[49]. Therefore, the increased abundance of TA genes may indicate intense gene content flux in the HL season, which mirrored that the abundance variation of TA genes were subjected to those in shared PCs. Specifically, the characterized leaf-response TA genes on the temperate phage genomes generally play roles in stringent response (*yefM-yoeB*)[50], multiple antibiotic resistance (*MarR*)[51], phage defence (*AbiEii* and *PemK*)[52], and growth inhibition of neighboring bacteria (Toxin-deaminase)[53], which are all proposed to confer eco-evolutionary benefits to the prokaryotic hosts. However, the increased abundance of *Zot* genes, which are prevalently found in pathogens[54], might suggest a susceptibility to disease for the gibbons in HL season.

Association analyses revealed that the phage-host community covariations were stronger for temperate phages than virulent types, which was accompanied with the shifts in the phage-host interaction structure. This finding contrasts results from a previous study where specialist phages showed higher correlations with their hosts than the generalist phages[55], as our analysis showed that virulent phages in the gibbon gut were more likely specialists while temperate phages tended to be generalists. This issue actually reflects the potential effect of

phage lifestyles on the strength of phage-host association, since temperate phages tend to coexist with their hosts, while virulent phages would lead to the lysis of host cells[56], which thus select for relatively broad and narrow host ranges, respectively. On the other side, the identification of significantly more of anti-phage systems for prokaryotes infected by virulent phages would indicate a higher resistance and thus narrower phage ranges. Furthermore, the larger number of phage defense systems detected may suggest a scenario under which a virulent phage population would strive to gain infectivity with either mutations in a given strain or multiple strain replacement within the population. This was corroborated by results of the comparative analysis of SNP profiles dynamics, which illustrated that virulent phages experienced more rapid loss of SNPs, resembling a series of selective purges[57]. Indeed, most SNPs on virulent and temperate genomes were found to be located in tail fiber genes involved in phage attachment to host cell surface and DNA primase that is essential for DNA replication, respectively[58,59]. Such patterns suggest that the evolution of virulent and temperate phages are mainly selected by the host cell-surface antiphage systems and replication machinery, respectively.

Despite subjected to limited sampling (six individuals only), our longitudinal study of the phages residing in the intestine of the skywalker hoolock gibbons provides initial insights into the complex nature of the gut phageome dynamics and its associations with prokaryotic hosts in wild animals with seasonal diet. Overall, the virulent and temperate types in the gut phageome responded differently to gibbon diet variations. The tight couplings in abundance, functions, and evolutionary dynamics between phages and their hosts highlight important roles of phages in influencing the eco-evolutionary dynamics of gut prokaryotes and eventually the adaptation of gibbons. Future investigations should ideally be integrated with more gibbon individuals or species, detailed nutritional ingredient analyses of the diet, and multi-omics to provide deeper insights into the ecological and evolutionary processes underlying patterns observed in field surveys. Such efforts will lead to a mechanistic understanding of how gut phage populations interact with their prokaryotic hosts in the context of natural diet variations and what are the functional and physiological consequences of such dynamic interactions. This will advance our knowledge of the evolution and adaptation of wild animals in their natural habitats, and ultimately contribute to the development of effective protection strategies for these endangered animal species on our planet.

## Methods

### Ethical approval for the monitoring of wild gibbons
We carried the field study and collected feces of wild skywalker hoolock gibbons (*H. tianxing*) at Nankang and Banchang in Mt. Gaoligong National Nature Reserve, Yunnan, P.R. China. All banked fecal samples were collected non-invasively during follows, and no disturbances or harms were caused by our collection. All data and samples were undertaken with permissions of Yunnan Gaoligongshan National Nature Reserve, Baoshan Bureau (permit date: 20161201).

### Wild gibbon populations and Data Collection
Skywalker hoolock gibbon was first described as a new species in 2017[60]. A latest survey showed that this species has a population of less than 150 individuals living in fragmented forests in Yunnan, China[61], thus being listed as Endangered in the IUCN Redlist[62]. We conducted our field study in two family groups (40 km apart) in Mt. Gaoligong National Nature Reserve, Yunnan, P.R. China: group NK (one adult male, one adult female, one infant, and one juvenile) at Nankang (24°49′N, 98°46′E) from October 2017 to October 2018, and group BC (one adult male and one adult female) at Banchang (25°12′N, 98°46′E) from October 2017 to December 2018. Before the study, both groups were well habituated to observers and all individuals were distinguished by their fur color, body size, and shape of the white eye

brow. We followed the adult female of each group (A2 in group NK and B2 in group BC) for on average eight days per month, each day from the time gibbons left their sleeping trees in the morning until they returned to the sleeping trees in the afternoon[63,64]. During follows, we recorded all feeding activities (5-min scan and ad libitum), recording the start time, end time, food species and food type (e.g., leaves, fruit, flower, animals, and others) of each feeding bout for the focal individual[25]. To calculate the dietary composition for each gibbon, the recorded feeding times allocated to the same food type were added and then divided by the total feeding time in a full-day. Once a defecation occurred, the landing location was immediately locked and 2–3 mL non-contaminated feces were collected within 5 min. Samples were sealed in 50 mL sterile tubes with 95% ethanol and then transported to the laboratory where they were stored at −80 °C[65]. To explore the potential impact of diet variaions on gut phages, only feces with parallel dietary data collected in the previous day were used in this study.

### Metagenomic sequencing and processing
Microbial DNA was extracted in September 2019 using the ALFA-SEQ Stool DNA Kit (Magigene, Guangdong, China) following the manufacturer's instructions. DNA concentration and purity were measured using Qubit 3.0 (ThermoFisher Scientific, Waltham, MA). A total of 138 microbial metagenomic DNA samples were used for library preparation with ALFA-SEQ DNA Library Prep Kit (Magigene, Guangdong, China) and sequenced from both ends on an Illumina NovaSeq 6000 platform (150 bp paired-end reads and 30 gb for each library). We also performed virus-like particles purification and DNA extraction for each fecal sample in August 2020 with minor modifications to the procedure described previously[66]. In brief, 0.5 g of each feces was suspended in a 3 mL 0.02-μm filtered saline magnesium buffer. After homogenization by vortexing for 5 min, the suspension was divided into pellet and supernatant by centrifugation at $2500 \times g$ for 10 min at 4 °C. We then resuspended the pellet with 2 mL saline magnesium buffer and repeated the above procedures to obtain complete separation of virus-like particles. All supernatants were collected and used for virus-like particles purification and DNA extraction[66]. Finally, 125 viral metagenomic DNA samples were obtained (with the other samples being discarded due to their low DNA yield/quality) and subsequently amplified using illustra™ Ready-To-Go™ GenomiPhi™ V3 DNA Amplification Kit (GE Healthcare, Amersham, UK). To estimate the degree of biases for the amplification, we randomly selected 10 viral metagenomic DNA samples for three replicates of amplification. Finally, a total of 145 amplified viral metagenomic DNA was then used for library preparation with NEBNext® Ultra II DNA Library Prep Kit for Illumina® (New England Biolabs, MA), and sequenced from both ends on an Illumina NovaSeq 6000 platform (150 bp paired-end reads and 15 gb for each library).

The viral and microbial metagenomic reads were trimmed and filtered using fastp v0.21.0 with default parameters[25,67]. For the high-quality metagenomic reads, superhost subtraction was performed against the *H. tianxing* genomes using Bowtie2 v2.2.9 to remove contaminant sequences from the gibbons[68]. The non-phage contamination and the degree to which enrichment for phage sequences was achieved in each VM or MM was evaluated by ViromeQC v1.0 (-w human)[69]. The clean reads for each metagenome were assembled independently using Megahit v1.2.9 and kmers of 21, 29, 39, 59, 79, 119, and 141[70].

### Identification and clustering of phage genomes
Phage genomes were identified from both the VMs and MMs assemblies using VirSorter v2.2.0 and CheckV v0.6.0[27,71]. Contigs longer than 10 kb were filtered with VirSorter with a minimal score of 0.95. The resultant dsDNA and ssDNA virus sequences were further processed with CheckV to trim potential host regions left at the ends of

prophages[27]. The output results from CheckV were used to screen potential phage contigs based on the following criteria: CheckV phage gene (≥1) or phage gene = 0 and (host gene = 0 or hallmark > 2). The output contigs were further manually checked through functional annotation. Specifically, Prodigal v2.6.3 and Prodigal v2.50 were run using the standard code (11) and three alternative genetic codes: TGA recoded tryptophan or glycine (code 4 or 25), TAG recoded to glutamine (code 15), and TAA recoded to glutamine (code 91)[72,73]. For contigs longer than 10 kb with GC content <50%, an alternative genetic code was predicted if its total coding score included in the GFF file was the highest and at least 10% higher than the standard genetic code[74]. The protein-coding sequences of recoded contigs were compared against the Virus Orthologous Group (release number: vog219, http://vogdb.org), and also proteins from crAss-like phages and Lak phages discovered previously since genes of these phages are extensively recoded[28,37,75,76]. The recoded contigs with at least one hallmark protein or ≥50% of annotated genes hit to a phage, combined with the output contigs with standard genetic code were regarded as putative phage genomes and used for subsequent analyses.

For the identified episomal phage genomes, Deephage v1.0 was used to distinguish virulent and temperate phage-derived sequences with default parameters[77]. Since the temperate phage is able to grow via both lytic and lysogenic replication pathways[26], the prophages identified by CheckV and episomal temperate phage genomes distinguished by Deephage were finally classified as temperate phages in each sample. All identified phage genomes that share 95% average nucleotide identity (ANI) and 85% alignment fraction were clustered into vOTUs based on the scripts (https://bitbucket.org/berkeleylab/checkv/src/master/) provided in CheckV[27]. Representative phage genomes were firstly taxonomically annotated using PhaGCN v2.0[78]. To specifically identify putative crAss-like phage and Lak phage, the representative phage genomes were searched against the published phage genomes of these two groups using BLASTn with an alignment longer than 1 kb and a minimum similarity of 90%[28,37,75,76]. Genomes in each vOTU were clustered into VM-vOTUs and MM-vOTUs based on the source of its members within the populations, or into virulent vOTUs (Vir-vOTUs) and temperate vOTUs (Tem-vOTUs) based on the predicted lifestyle of its members within the populations. To generate abundance profiles across samples, reads from the 145 VMs libraries were aligned to the representative genomes of VM-vOTUs and Vir-vOTUs, while reads from the 138 microbial metagenomes were aligned to the representative genomes of MM-vOTUs and Tem-vOTUs. The reads alignment was performed using BamM v1.7.3 (http://ecogenomics.github.io/BamM/) with default parameters, and the coverage of each genome was calculated using the 'tpmean' coverage mode (remove the highest 5% and the lowest 5% coverage regions, minimum nucleotide identity of 95%, minimum aligned length of 75% of each read)[79].

## Phage-host linkages

We firstly recovered the prokaryotic genomes from the MMs. The assembled contigs from each MM were binned to draft-quality genomes using MetaBAT v2.12.1 considering both tetranucleotide frequency and reads abundance[80]. All bins were then curated to obtain high-quality genomes using RefineM v0.0.24[81]. We also removed the 'co-binned' phage contigs from the genomes. The completeness and contamination of these curated genomes were estimated using CheckM2 v1.0.2[82]. The resulted genomes were further aggregated and clustered into species-level using dRep v3.2.2 with options 'dereplicate -comp 50 -con 10 -sa 0.95 -cm larger -nc 0.3'[83,84]. The 597 dereplicated genomes were taxonomically annotated using the genome taxonomy database (GTDB-Tk v2.3.0)[85]. Reads from each of the 138 MMs were mapped to the set of dereplicated genomes using BamM v1.7.3 (http://ecogenomics.github.io/BamM/) with default parameters, and the coverage of each genome was calculated using the 'tpmean' coverage mode (remove the highest 5% and the lowest 5% coverage regions,

minimum nucleotide identity of 95%, minimum aligned length of 75% of each read)[79].

We used a combination of Clustered Regularly Interspaced Short Palindromic Repeats (CRISPRs)-spacer matches and genome sequence matches to link phage genomes to prokaryotic genomes. The software metaCRT v1.2 was used to predict CRISPR spacers from 10,567 recovered prokaryotic genomes (≥50% completeness and ≤10% contamination) with default parameters[86]. The spacer sequences were collected and compared to phage genomes using BLASTn with thresholds of an E-value ≤ $10^{-10}$ and no mismatches over the whole spacer length. Besides, all recovered phage genomes and the 10,567 prokaryotic genomes (completeness ≥50% and contamination <10%) were compared using BLASTn with a criteria of E-value ≤ $10^{-3}$, bit score ≥50, alignment length ≥1 kb and identity ≥96%[74].

## Protein clustering and functional annotation

To decontaminate proteins originated from phages, we removed contigs with prophages in prokaryotic genomes. The protein-coding genes on these trimmed prokaryotic genomes were then predicted by Prodigal v2.6.3 with the parameters set as '-p meta -g 11 -f gff -q -m'[72]. The predicted prokaryotic proteins, combined with the predicted protein sequences from virulent and temperate phage genomes were clustered into PCs using cd-hit v4.8.1 with parameters of '-n 4 -d 0 -g 1 -c 0.6. -aS 0.8'[87]. The abundance of a phage PC across samples were determined as the total abundance of phage populations encoding proteins in that PC. This could avoid the potential effects of host-derived reads on the abundances of phage PCs and ensure that the phage PCs was explicitly derived from phages[47]. The proportion of shared PCs for each phage-host pair were firstly calculated as the number of shared PCs between them divided by the total number of phage PCs. We also calculated the proportion of shared PCs between the same set of phages and the nonhosts in each prokaryotic family. Since the number of phage-nonhost pairs were larger than the number of phage-host pairs in each family, the proportion of shared PCs calculated between phage-nonhost pairs were subsampled to ensure an equal number of pairs for hosts and nonhosts in each family. The specific scripts used to generate Fig. 4d, e are now stored in figshare as described in Code availability section. To determine the anti-phage defense genes on prokaryotic genomes, the predicted prokaryotic genes were annotated using Defense-Finder v1.2.0[88]. The unique protein-coding gene in each PC was annotated based on HMM searches against Pfam database (release number: Pfam36.0) and the VOG database (release number: vog219, http://vogdb.org) using the hmmsearch v3.3.2 (threshold of 50 for bit score and $10^{-5}$ for E-value)[89,90].

## SNPs calling

Virulent and temperate population genomes with a minimum sequencing depth of 10-fold across all samples of each gibbon were flagged for calling SNPs[91]. In brief, reads were mapped to the phage genomes independently per sampling date as above. Samtools v1.17 and bcftools v1.17 were used to identity and filter variations among reads per site that had a quality call >30 and occur in at least 4 reads[91]. Meanwhile, the alternative alleles that had a frequency >1% were only considered to be bona fide and used for downstream analysis[91,92]. The SNP profile at each date was then compared to all of the subsequent dates. The proportion of shared SNPs between two dates was calculated as the total number of shared SNPs divided by the total number of SNPs identified in the previous date. We employed the SNPEff v5.2 software to identify non-synonymous and synonymous substitutions[93].

## Statistical analyses

Statistical analyses were performed with various packages in the statistical program R v4.0.3[94]. Phage and prokaryotic abundance matrices

were standardized using 'decostand' function in vegan v2.5-5 with methods of 'Hellinger'[95]. PCoA (utilizing the Bray-Curtis dissimilarity metric) was used (vegan v2.5-5) to understand how phage communities vary between individuals. PERMANOVA (999 permutations) was performed using 'adonis' function in vegan v2.5-5 to test for significant differences in phage community composition and dietary composition between samples[95]. The Shapiro-Wilk test and Bartlett's test were used to check for normality and homoscedasticity between groups, respectively. Statistical significance of differences was then determined using non-parametric Wilcoxon *t* test (unpaired). *P* values for multiple testing were adjusted using the Benjamini and Hochberg false discovery rate controlling procedure (stats v4.0.3)[96]. Pearson's correlations were calculated using 'rcorr' function in Hmisc v4.2-0 to assess the correlations[97]. Relationships between different factors across samples were investigated by linear regression analyses using 'lm' function in vegan v2.5-5[95].

### Reporting summary

Further information on research design is available in the Nature Portfolio Reporting Summary linked to this article.

## Data availability

Raw viral metagenomic sequencing data have been deposited in NCBI BioProject database under accession code PRJNA831632. Microbial metagenomic sequencing data have been deposited in the National Genomics Data Center, China National Center for Bioinformation or Beijing Institute of Genomics, Chinese Academy of Sciences under the accession code PRJCA012504. Biosample accession numbers for individual microbial metagenomes and viral metagenomes are listed in Supplementary Data 11. The raw data for phage and prokaryotic richness, taxonomic and functional composition, genomic features, protein clusters, and other relevant data used to generate figures for this study are available from https://doi.org/10.6084/m9.figshare.20099621. Source data are provided with this paper.

## Code availability

The in-house R scripts and relevant data used to generate figures for this study are provided with this paper and publicly available from https://doi.org/10.6084/m9.figshare.20099621.

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

## Acknowledgements

We would like to thank Mt. Gaoligong National Nature Reserve for assistance with field work, Li Yang for assistance with the plotting of maps. This work was funded by the Ministry of Science and Technology of China: National Key Research and Development Program (grant no. 2022YFF1301500 to P.F.F.), the National Natural Science Foundation of China (nos. 31822049 and 31770421 to P.F.F., no. 32201269 to S.M.G.), the Guangdong Basic and Applied Basic Research Foundation (nos. 2021A1515110523 and 2023A1515010695 to S.M.G.), the China Postdoctoral Science Foundation (no. 2021M703754 to S.M.G.), and Fundamental Research Funds for the Central Universities, Sun Yat-sen University (no. 23lgzy002 to P.F.F., no. 22qntd2617 to S.M.G.).

## Author contributions

Conceptualization, P.F.F.; Methodology, P.F.F. and L.N.H.; Sampling, H.L.F.; Experiments, S.M.G., Q.L. and L.Y.L.; Formal analysis, S.M.G.; Writing original draft, S.M.G.; Review and editing, P.F.F. and L.N.H.; Supervision, P.F.F. and L.N.H.; Funding acquisition, P.F.F. and S.M.G. All authors read and approved the final manuscript.

## Competing interests

The authors declare no competing interests.
