## [Peer Review File · Nature Communications]

REVIEWER COMMENTS

Reviewer #1 (Remarks to the Author):

Here are some concerns.

- (1) The sample size of the individual is small and only includes six individuals. Thus, how did the author conclude strongly based on this small sample size?
- (2) The quality of fecal samples and DNAs. The authors mentioned these samples were collected from 2017 to 2018. Thus, when did the authors make the DNA extraction? Now, the date is 2023. I worry about the DNA quality. The authors should give us or readers this basic information.
- (3) These are two common metagenomics methods, and the manuscript needs more novelty if only comparing these different methods.
- (4) The authors should be cautious about the conclusion. This study only uses DNA, not RNA. Thus, most of the viruses come from phages, and there is no information on the RNA virus. Therefore, how could the authors talk about the gut virome of the gibbon?
- (5) Also, the study only includes one gibbon species. Thus, the title should be clear.

Reviewer #2 (Remarks to the Author):

The manuscript by Gao et al aims at characterizing the viral communities of two female gibbon individuals sampled in ~50 occasions over a 1-year period. They used two complementary approaches including VLP enriched and bulk metagenomes and compared the obtained viral populations, associations with diet changes and ecological dynamics.

Is a complex and well written manuscript, although several typos and complex sentences were seen, that could benefit of a thorough review. That uses state of the art methods and benefits greatly from a deep sampled and deep sequenced individuals. The results are interesting and relevant although mostly they confirm observations that have already been made in other primates (including humans).

I only have few main comments and several minor comments:

- 1) When deriving conclusions from data analysis of massive datasets such as this, I'm usually cautious and try to make the reader aware of the biases and limitations that the methods can have. In particular, error propagation could be something quite significant. What I mean is that usually each software used will incorporate some bias, due to training limitations or parameters used, for example, DNA extraction leaves out all the RNA viruses, then assembly will limit only highly abundant phages, then a size selection is done removing important small viruses, then checkV is used removing potentially new viruses that do not fit on the known characteristics of viruses. Those cumulative biases may reduce the final dataset to one that may not represent what could be happening with the whole virome. Those limitations should be mentioned.

2) Although is very interesting the identification of prophages from the bulk metagenome, and although I understand the prediction of temperate vs virulent is done computationally, it is risky to label them as temperate since by definition, a temperate virus is one that could be induce, otherwise is a prophage that could potentially be a dead virus with no implication for viral and ecological dynamics. The set of phages that are only seen in the bulk metagenome but not in the VLP one, how do you know if they can be induced? And if not, what utility may those have? They are just simply reflecting the abundance of the host where they are located, they may even be false positive calls of the viral identification software.

3) The phenomenon observed on substitution rates is interesting, would it be different if it is split by viral taxonomy? Different viral families may have different substitution rates and if a family is unique or has higher abundance in an individual than the other one may cause different (contrasting) results as observed for the two individuals. The shape of the density plots can also be due to the low number of phages where substitution rates could be measured, this should also be mentioned and highlighted. Related, is very interesting the observation that for both individuals and both types of viruses there is a decrease in substitution rate with time. This is not discussed and seem counterintuitive for me, I would expect less divergence between viruses that are closer in time than those that have been evolving for longer time, or am I misinterpreting the data?

4) An observation from figure 2b is quite interesting, where there is a higher amount of low prevalence bulk specific viruses but with higher relative abundance. What can be causing that peak at middle prevalence not observed on the VLP enriched specific viruses?

Minor comments

Line 20: We attempt to what? Collect? Process, analyze? Characterize?

Line 429, it highlight that sequences of more than 10Kb were not analyzed, but is not clear if they were removed from all analyses or just a subset. There is a large range of microviridae in the 4-6Kb range that would have been removed.

At some point it is explained that 10 samples were used for triplicate MDA, thus reaching 110 sequenced libraries from 90 DNA samples, however that is after citing Fig 1b and Supplementary Data 1, which can generate some confusion. It should be ideally made clear before that.

Figure 3b. What does it means shared? It is predicted as virulent and temperate? By definition it is not possible to be both.

Data S8 will be ideal to point whether the match was done via CRISPR spacer or shared genomic matches, in cases of host genomic matches on virulent viruses it will be important to explain why this could happen given that the virulent phages should not integrate in the genome.

Figure 4c, the correlation seems mainly driven by a few outliers, it will be important to do a normality test before the regression or do an outlier removal before the test.

Fig 5c, how the modularity and nestedness relates with the co-evolutionary dynamics may deserve a further explanation.

Both in the introduction and the methods an specific emphasis was done on mapping to crassphage and lakphages but no results on those is shown.

Reviewer #3 (Remarks to the Author):

The article by Gao and colleagues describes the viral metagenomes and microbial metagenomes obtained from two individual wild gibbons over a period of 15 month, focusing primarily on the comparison of the two metagenome approaches and reporting significant differences between them. While I am not sure if there are any papers systematically comparing these two approaches, I believe that this study was not designed for this neither; if it had, it would have included additional type of samples (as opposed to a single type of sample currently) and a couple control samples (mock samples). Further, it is common to compare abundances between the cellular and viral fractions, and there is usually an inverse correlation (i.e., actively infecting viruses are not abundant in the virome and vice versa), exactly what this study reports. See specific publications for this:

<https://doi.org/10.1128/msystems.00905-19> (see table 1 and figure 4)

<https://doi.org/10.1111/1462-2920.15510> (see table 2)

There is even a recent review that also talks about these two approaches to identify viruses (see Box 1):

<https://www.ncbi.nlm.nih.gov/pmc/articles/PMC7157462/>

So, in the view of this reviewer, the current focus of the paper is not novel enough and should not be the main focus of the manuscript. The manuscript should be refocused on the biological insights and, in fact, outline a couple hypotheses in the introduction that are tested later with the data as opposed to just listing results, which is the current style. The dataset is great and will be used by several researchers in the future, I anticipate, but it is clearly underutilized in this study. The authors could get into the pressing eco-evolutionary dynamics between viruses and their hosts questions, for example.

The bioinformatics analyses look good overall but there is some lack of details in describing the samples used and lack of discussion of related literature (a lot of literature is cited but not always as it relates to the actual results reported; as an example, see above that the paper did not really discuss the result of the previous studies that compared viral and microbial metagenomes). The manuscript requires substantial editorial editing for its language. I mentioned some representative examples from only the abstract below; reporting all grammatical inaccuracies and typos is not practical.

Minor/specific points:

Ln 20. Sequenced?

Ln 24. WERE biased...(in capital letters, the word missing).

Ln 26. What does it mean a virus is more closely related to a host? Maybe say instead "tighter association"?

Ln 29. Do you mean "gibbon individual-specific"? Or what exactly? Also, how base substitutions differed really (e.g., nonsynonymous vs synonymous substitutions)? Please specify.

Ln 30-32. Seems like a trivial conclusion based on existing literature. Can the authors come up with something more novel, maybe from the actual biological (not technical as done here) results?

Ln 49. Small genome size or small cell size? The two do not go together necessarily. Also, maybe it is preferable to refer to bulk MICROBIAL metagenomes; and subsequently just microbial metagenomes (or cellular vs. viral fraction as mentioned above)? Bulk metagenomes seems awkward to me at least.

Ln 55. Increasing or enhancing? (Instead of maximizing)

Ln 78-79. Are the two methods applied to the same exact samples or not? Also, if the study was indeed to compare the two methods, a wider range of samples should have been tested, including a couple samples of known composition (mock samples). So, the motivation for the study, as stated in the Introduction section, is not very strong. It will be probably better to spin this manuscript around the daily and diet changes in the gut microbiome of the gibbons and/or the dynamics between the viruses and their hosts in my view.

Ln 126. Based on what criteria was completeness assessed? Mention briefly in the main text so it can stand on its own.

Ln 139-140. Why minor if clustered separately (seems more important than the authors imply here)? Also, need to describe the samples a bit better; how many replicates, were the two methods applied to all samples, etc. Need also to mention what fraction of the viral genomes the authors were able to assign a host to (applies later in the manuscript).

Ln 145. Say by how much to make text stand on its own, independent of the figures.

Ln 154-155. Why unexpected? Clarify?

Ln 163-165. Based on what criteria? Just being present in the microbial metagenomes or another type of analysis? Please specify.

Ln 251-253. How you are sure these are base substitution differences as opposed -for example- to population with higher intra-population diversity and/or more frequent strain replacement? Also, what substitutions there were e.g., mostly neutral or non-synonymous indicating selection-driven mutations? These are interesting questions biologically to report on.

Ln 307. Individuals not groups really, right?

Reviewer #1 (Remarks to the Author):

Here are some concerns.

(1) The sample size of the individual is small and only includes six individuals. Thus, how did the author conclude strongly based on this small sample size?

Response: We acknowledge that the number of individuals sampled in our study is small. This limitation is now stated in the concluding remarks (Lines 438 and 446), and conclusive statements are weakened accordingly in the manuscript. For wild animals, quantifying diet and collecting individually recognized fresh feces for consecutive days face multiple challenges (Lines 73-85). Our work was even more difficult as the skywalker hoolock gibbon is a tree-dwelling species, and it is endangered with less than 150 individuals living in fragmented forests in Yunnan Province, China (Lines 458-461). Thus, we focused on all six members in two gibbons family groups over consecutive 15 months in the current study.

(2) The quality of fecal samples and DNAs. The authors mentioned these samples were collected from 2017 to 2018. Thus, when did the authors make the DNA extraction? Now, the date is 2023. I worry about the DNA quality. The authors should give us or readers this basic information.

Response: All fecal samples were stored at -80°C prior to DNA extraction. The DNAs for total metagenomes and viral metagenomes were extracted in September 2019 and August 2020, respectively. This information is now added in the Methods section (Lines 483 and 491). DNA yields and quality were analyzed, and those with low yields or quality were discarded (Lines 499).

(3) These are two common metagenomics methods, and the manuscript needs more novelty if only comparing these different methods.

Response: Thank you for your constructive suggestions. We have refocused on the biological findings, namely the dynamics of phage communities in response to the seasonal diet variations of gibbons and its associations with prokaryotic hosts. As a result, the Introduction, Results, and Discussion sections are substantially rewritten and improved.

(4) The authors should be cautious about the conclusion. This study only uses DNA, not RNA. Thus, most of the viruses come from phages, and there is no information on the RNA virus. Therefore, how could the authors talk about the gut virome of the gibbon?

Response: We now use the word 'phage' or 'phageome' instead of 'virus' and 'virome' throughout the manuscript.

(5) Also, the study only includes one gibbon species. Thus, the title should be clear.

Response: The species name (*Hoolock tianxing*) of our studied gibbon is now added in the title.

Reviewer #2 (Remarks to the Author):

The manuscript by Gao et al aims at characterizing the viral communities of two female gibbon individuals sampled in ~50 occasions over a 1-year period. They used two complementary approaches including VLP enriched and bulk metagenomes and compared the obtained viral populations, associations with diet changes and ecological dynamics.

Is a complex and well written manuscript, although several typos and complex sentences were seen, that could benefit of a thorough review. That uses state of the art methods and benefits greatly from a deep sampled and deep sequenced individuals. The results are interesting and relevant although mostly they confirm observations that have already been made in other primates (including humans).

I only have few main comments and several minor comments:

1) When deriving conclusions from data analysis of massive datasets such as this, I'm usually cautious and try to make the reader aware of the biases and limitations that the methods can have. In particular, error propagation could be something quite significant. What I mean is that usually each software used will incorporate some bias, due to training limitations or parameters used, for example, DNA extraction leaves out all the RNA viruses, then assembly will limit only highly abundant phages, then a size selection is done removing important small viruses, then checkV is used removing potentially new viruses that do not fit on the known characteristics of viruses. Those cumulative biases may reduce the final dataset to one that may not represent what could be happening with the whole virome. Those limitations should be mentioned.

Response: We thank you for your reminding. Clarifications have been made in the new version of manuscript to remind the readers of the limitations. We use the word 'phage' or 'phageome' instead of 'virus' and 'virome' throughout the manuscript. We have analyzed and discussed about the potential biases of the two metagenomic methods in Lines 164-168 and 347-350. Limitations of bioinformatic analyses are also highlighted in Lines 350-355. Additionally, we acknowledge in the concluding remarks that the number of gibbon individuals sampled in our study is small due to challenges in quantifying diet and collecting individually recognized fresh feces for consecutive days for these tree-dwelling, endangered animals (Lines 438 and 446).

2) Although is very interesting the identification of prophages from the bulk metagenome, and although I understand the prediction of temperate vs virulent is done computationally, it is risky to label them as temperate since by definition, a temperate virus is one that could be induce, otherwise is a prophage that could potentially be a dead virus with no implication for viral and ecological dynamics. The set of phages that are only seen in the bulk metagenome but not in the VLP one, how do you know if they can be induced? And if not, what utility may those have? They are just simply reflecting the abundance of the host where they are located, they may even be false positive calls of the viral identification software.

Response: It is true that temperate phages would be integrated into host genome (i.e., prophage) or be episomal in host cells (Correa et al., 2021, <https://doi.org/10.1038/s41579-021-00530-x>). Nevertheless, we acknowledge that currently it is challenging to distinguish whether the prophage is active and thus could be induced. However, the prophage can also function by contributing to the physiology (Wang et al., 2010, <https://doi.org/10.1038/ncomms1146>), competitive advantages (Duerkop et al., 2012, <https://doi.org/10.1073/pnas.1206136109>), or gene expression (Feiner et al., 2015, <https://doi.org/10.1038/nrmicro3527>) of their microbial hosts. Therefore, we classified the prophage as the temperate phage, as the procedures widely used in recent viral ecology studies (Brum et al., 2016, <https://doi.org/10.1038/ismej.2015.125>; Morris et al., 2019, <https://doi.org/10.1038/s41564-020-0725-x>; Luo et al., 2022, <https://doi.org/10.1186/s40168-022-01384-y>). This is now clarified in Lines 540-543. Additionally, our analyses showed that temperate phage, including the prophage, contained toxin-antitoxin genes responsive to season diets in the guts of skywalker hoolock gibbons (Lines 273-288), implying the potential roles of prophages in mediating the metabolism of their microbial hosts (Lines 401-414).

3) The phenomenon observed on substitution rates is interesting, would it be different if it is split by viral taxonomy? Different viral families may have different substitution rates and if a family is unique or has higher abundance in an individual than the other one may cause different (contrasting) results as observed for the two individuals. The shape of the density plots can also be due to the low number of phages where substitution rates could be measured, this should also be mentioned and highlighted. Related, is very interesting the observation that for both individuals and both types of viruses there is a decrease in substitution rate with time. This is not discussed and seem counterintuitive for me, I would expect less divergence between viruses that are closer in time than those that have been evolving for longer time, or am I misinterpreting the data?

Response: The phenomenon that the base substitution rates decrease as the measurement timescales increase has been reported in previous studies and is possibly driven by several factors, i.e., natural selection, substitution saturation, etc (Ho et al., 2011, <https://doi.org/10.1111/j.1365-294x.2011.05178.x>; Aiewsakun and Katzourakis, 2016, <https://doi.org/10.1128/jvi.00593-16>). While this phenomenon was also observed for virulent and temperate phages in our previous draft, we do not include the calculation and comparison of the base substitution rates for the virulent and temperate phages in the revised manuscript. The base substitution rate is calculated as the mean proportion of bases that are different across every date normalized for the genome sequencing depth (Minot et al., 2013; <https://doi.org/10.1073/pnas.1300833110>). We observed an order of magnitude higher of sequencing depth for the virulent phages than temperate phages (Lines 317-318, Supplementary Fig. 15). This might introduce biases in SNPs calling that could not be thoroughly eliminated by the sequencing depth normalization, as the number of called SNPs is not linearly correlated with sequencing depth (Kishikawa et al., 2019,

<https://doi.org/10.1038/s41598-018-38346-0>). Therefore, in this new version of manuscript, we only focused on the proportion of shared SNPs between dates which could reflect the overall genomic similarity across the time lags to avoid using of sequencing depth (Fig. 5f, Lines 320-331).

4) An observation from figure 2b is quite interesting, where there is a higher amount of low prevalence bulk specific viruses but with higher relative abundance. What can be causing that peak at middle prevalence not observed on the VLP enriched specific viruses?

Response: We have now compared (Lines 164-168) and discussed (Lines 347-350) the phage prevalence observed in viral and total metagenomes.

Minor comments

Please note that some of the minor comments raised below might not be relevant (e.g., related parts/items are deleted) in the revised manuscript due to a shift of the focus as suggested by other two Reviewers.

Line 20: We attempt to what? Collect? Process, analyze? Characterize?

Response: This is now revised in Line 23.

Line 429, it highlight that sequences of more than 10Kb were not analyzed, but is not clear if they were removed from all analyses or just a subset. There is a large range of microviridae in the 4-6Kb range that would have been removed.

Response: This is discussed in Lines 350-355.

At some point it is explained that 10 samples were used for triplicate MDA, thus reaching 110 sequenced libraries from 90 DNA samples, however that is after citing Fig 1b and Supplementary Data 1, which can generate some confusion. It should be ideally made clear before that.

Response: This is clarified in Lines 109-111, and in Fig.1b.

Figure 3b. What does it mean shared? It is predicted as virulent and temperate? By definition it is not possible to be both.

Response: This is clarified in Lines 180-182.

Data S8 will be ideal to point whether the match was done via CRISPR spacer or shared genomic matches, in cases of host genomic matches on virulent viruses it will be important to explain why this could happen given that the virulent phages should not integrate in the genome.

Response: The host prediction method for each phage-host pair is now added in Supplementary Data 8. Additionally, while virulent phages would not integrate into host genomes, they might contain genes horizontally transferred from their hosts, which could contribute to the phage-host genomic matches (Luo et al., 2022, <https://doi.org/10.1186/s40168-022-01384-y>).

Figure 4c, the correlation seems mainly driven by a few outliers, it will be important to do a normality test before the regression or do an outlier removal before the test.

Response: This figure is deleted.

Fig 5c, how the modularity and nestedness relates with the co-evolutionary dynamics may deserve a further explanation.

Response: The results related to modularity and nestedness are not included in the revised manuscript.

Both in the introduction and the methods an specific emphasis was done on mapping to crAssphage and lakphages but no results on those is shown.

Response: The crAss-like phage and Lak phage are now analysed and discussed in Lines 182-187, 350-355, and 547-550.

Reviewer #3 (Remarks to the Author):

The article by Gao and colleagues describes the viral metagenomes and microbial metagenomes obtained from two individual wild gibbons over a period of 15 month, focusing primarily on the comparison of the two metagenome approaches and reporting significant differences between them. While I am not sure if there are any papers systematically comparing these two approaches, I believe that this study was not designed for this neither; if it had, it would have included additional type of samples (as opposed to a single type of sample currently) and a couple control samples (mock samples). Further, it is common to compare abundances between the cellular and viral fractions, and there is usually an inverse correlation (i.e., actively infecting viruses are not abundant in the virome and vice versa), exactly what this study reports. See specific publications for this:

<https://doi.org/10.1128/msystems.00905-19> (see table 1 and figure 4)

<https://doi.org/10.1111/1462-2920.15510> (see table 2)

There is even a recent review that also talks about these two approaches to identify viruses (see Box 1):

<https://www.ncbi.nlm.nih.gov/pmc/articles/PMC7157462/>

So, in the view of this reviewer, the current focus of the paper is not novel enough and should not be the main focus of the manuscript. The manuscript should be refocused on the biological insights and, in fact, outline a couple hypotheses in the introduction that are tested later with the data as opposed to just listing results, which is the current style. The dataset is great and will be used by several researchers in the future, I anticipate, but it is clearly underutilized in this study. The authors could get into the pressing eco-evolutionary dynamics between viruses and their hosts questions, for example.

The bioinformatics analyses look good overall but there is some lack of details in

describing the samples used and lack of discussion of related literature (a lot of literature is cited but not always as it relates to the actual results reported; as an example, see above that the paper did not really discuss the result of the previous studies that compared viral and microbial metagenomes). The manuscript requires substantial editorial editing for its language. I mentioned some representative examples from only the abstract below; reporting all grammatical inaccuracies and typos is not practical.

Response: We thank you for your constructive comments. These major issues have been carefully addressed in the new version of manuscript. Specifically, hypotheses have been made in the Introduction (Lines 94-101) and tested in the Results section (Lines 243-244 and 287-288). The new manuscript refocuses on the response of gibbon gut phageome to seasonal diet variations (Fig. 3 and 4), and the phage-host eco-evolutionary dynamics (Fig. 5). We have clarified in Lines 109-111 and in Fig. 1a, b about our samples and the datasets. More relevant literature, including the above mentioned papers, are cited and discussed with our results (e.g., in Lines 350 and 357). A thorough language editing of the manuscript is conducted, with a specific focus on complex sentences and grammatical mistakes, etc. Please note that some of the minor/specific comments raised below might not be relevant (e.g., related parts/items are deleted) in the new manuscript due to a shift of the focus.

Minor/specific points:

Ln 20. Sequenced?

Response: This is now revised in Line 23.

Ln 24. WERE biased...(in capital letters, the word missing).

Response: The abstract section is now substantially revised, and this sentence is deleted.

Ln 26. What does it mean a virus is more closely related to a host? Maybe say instead “tighter association”?

Response: This sentence is deleted.

Ln 29. Do you mean “gibbon individual-specific”? Or what exactly? Also, how base substitutions differed really (e.g., nonsynonymous vs synonymous substitutions)? Please specify.

Response: This sentence is deleted.

Ln 30-32. Seems like a trivial conclusion based on existing literature. Can the authors come up with something more novel, maybe from the actual biological (not technical as done here) results?

Response: This conclusion sentence in the abstract is revised (Lines 37-39).

Ln 49. Small genome size or small cell size? The two do not go together necessarily. Also, maybe it is preferable to refer to bulk MICROBIAL metagenomes; and

subsequently just microbial metagenomes (or cellular vs. viral fraction as mentioned above)? Bulk metagenomes seems awkward to me at least.

Response: The introduction section is substantially revised, and this sentence is deleted. Meanwhile, the ‘bulk metagenome’ is now replaced with ‘total metagenome’ throughout the manuscript.

Ln 55. Increasing or enhancing? (Instead of maximizing)

Response: This sentence is deleted.

Ln 78-79. Are the two methods applied to the same exact samples or not? Also, if the study was indeed to compare the two methods, a wider range of samples should have been tested, including a couple samples of known composition (mock samples). So, the motivation for the study, as stated in the Introduction section, is not very strong. It will be probably better to spin this manuscript around the daily and diet changes in the gut microbiome of the gibbons and/or the dynamics between the viruses and their hosts in my view.

Response: The detailed sampling information and our metagenomic datasets are now described in Lines 109-111 and also provided in Fig. 1a, b. Meanwhile, we have repositioned the main focus of our manuscript on the eco-evolutionary dynamics of phage communities and its associations with gibbons’ seasonal diet variations and prokaryotic hosts.

Ln 126. Based on what criteria was completeness assessed? Mention briefly in the main text so it can stand on its own.

Response: This is mentioned as suggested (Lines 130-131).

Ln 139-140. Why minor if clustered separately (seems more important than the authors imply here)? Also, need to describe the samples a bit better; how many replicates, were the two methods applied to all samples, etc. Need also to mention what fraction of the viral genomes the authors were able to assign a host to (applies later in the manuscript).

Response: This sentence is revised (Lines 145-148). The detailed information for our samples and datasets are provided in Lines 109-111 and in Fig. 1a, b. Also, the fraction of phages assigned to a host is added in Line 219.

Ln 145. Say by how much to make text stand on its own, independent of the figures.

Response: The detailed value is provided in the main text (Lines 151-155).

Ln 154-155. Why unexpected? Clarify?

Response: This is clarified in Lines 161-164.

Ln 163-165. Based on what criteria? Just being present in the microbial metagenomes or another type of analysis? Please specify.

Response: The phage lifestyles were predicted by using the Deephage software as

described in Lines 538-543.

Ln 251-253. How you are sure these are base substitution differences as opposed -for example- to population with higher intra-population diversity and/or more frequent strain replacement? Also, what substitutions there were e.g., mostly neutral or non-synonymous indicating selection-driven mutations? These are interesting questions biologically to report on.

Response: It is true that the strain replacement within populations could also contribute to the base substitution differences between dates. This is now discussed in Lines 426-429. Meanwhile, the substitution types (nonsense, missense, and silent) are now analyzed and discussed in Lines 326-331 and 432-437.

Ln 307. Individuals not groups really, right?

Response: This sentence is deleted.

REVIEWERS' COMMENTS

Reviewer #1 (Remarks to the Author):

I appreciate the authors' effort in the revision. However, I still worry about the small sample size and lack of novels. Also, this study's other limitation is only having the DNA virus information. Thus, this is not a well-designed study.

Reviewer #3 (Remarks to the Author):

The revised article by Gao et al., represents a much-improved version compared to the first submission. The authors have addressed all my major concerns and those of other reviewers, including the suggestion to remove emphasis on method comparison and focus more on the biological results and interpretations. I felt the authors did a good job overall integrating reviewers' input and revising the manuscript. I have no major comments remaining. I have only a few minor issues for the authors to consider toward further clarifying their manuscript.

Abstract, line 15. 15 CONSECUTIVE months (not consecutive 15 months). Some editorial editing is still required (a representative examples provided here and about line 299 below).

Abstract, lines 35-37. Can the authors come up with something more novel/exciting to replace this sentence with? e.g. this same sentence is repeated in the introduction section for the background information almost intact. Maybe to say something about selected functions based on SNP patterns? Or what type of viruses or which viral families different diets seem to select for? (a couple examples to consider)

Also, why not use virome which is more widespread, I think, compared to phageome?

I don't think Total Metagenome -TM- is a better term. First, total is never achieved really; and secondarily, the microbial metagenome undersamples the viromes as the authors show in this study too (so it is not "total"). Not sure why the authors want to come up with new or unusual terms when accurate enough and widespread terms already exist for the same things.

Results. I like the summary of TM vs VM comparisons (i.e. the change in focus compared to original version), albeit the comparison probably takes a bit too long and could be shortened somewhat, I felt.

Lines 299-300. Might correspond to (not be corresponded).

REVIEWERS' COMMENTS

Reviewer #1 (Remarks to the Author):

I appreciate the authors' effort in the revision. However, I still worry about the small sample size and lack of novels. Also, this study's other limitation is only having the DNA virus information. Thus, this is not a well-designed study.

Response: Thanks. It often takes years to habituate one gibbon group in China. We will certainly include more samples/gibbon individuals, and both DNA and RNA viruses in our future studies.

Reviewer #3 (Remarks to the Author):

The revised article by Gao et al., represents a much-improved version compared to the first submission. The authors have addressed all my major concerns and those of other reviewers, including the suggestion to remove emphasis on method comparison and focus more on the biological results and interpretations. I felt the authors did a good job overall integrating reviewers' input and revising the manuscript. I have no major comments remaining. I have only a few minor issues for the authors to consider toward further clarifying their manuscript.

Response: Thank you very much for your positive comments.

Abstract, line 15. 15 CONSECUTIVE months (not consecutive 15 months). Some editorial editing is still required (a representative examples provided here and about line 299 below).

Response: We have now corrected grammatical mistakes throughout the manuscript (Lines 19 and 282).

Abstract, lines 35-37. Can the authors come up with something more novel/exciting to replace this sentence with? e.g. this same sentence is repeated in the introduction section for the background information almost intact. Maybe to say something about selected functions based on SNP patterns? Or what type of viruses or which viral families different diets seem to select for? (a couple examples to consider)

Response: This sentence is now rephrased as suggested (Lines 29-31).

Also, why not use virome which is more widespread, I think, compared to phageome?

Response: We use the term 'phageome' since there is no information on the RNA virus in our study as Reviewer #1 suggested in the previous round of review.

I don't think Total Metagenome -TM- is a better term. First, total is never achieved really; and secondarily, the microbial metagenome undersamples the viromes as the authors show in this study too (so it is not "total"). Not sure why the authors want to come up with new or unusual terms when accurate enough and widespread terms already exist for the same things.

Response: We replace the 'total metagenome (TM)' with 'microbial metagenome (MM)' throughout the manuscript text and figures as suggested.

Results. I like the summary of TM vs VM comparisons (i.e. the change in focus compared to

original version), albeit the comparison probably takes a bit too long and could be shortened somewhat, I felt.

Response: The section comparing microbial metagenomes and viral metagenomes are now shortened with the results unchanged (Lines 132-176).

Lines 299-300. Might correspond to (not be corresponded).

Response: This is now revised (Line 282).